# Factors Affecting the Parental Intention of Using AVs to Escort Children: An Integrated SEM–Hybrid Choice Model Approach

**Yueqi Mao [1], Qiang Mei [1,\*], Peng Jing [2,\*], Ye Zha [2], Ying Xue [2], Jiahui Huang [2], Danning Shao [2] and Pan Luo [2]**

1   School of Automotive and Traffic Engineering, Jiangsu University, Zhenjiang 212013, China
2   School of Management, Jiangsu University, Zhenjiang 212013, China
\*   Correspondence: qmei@ujs.edu.cn (Q.M.); jingpeng@ujs.edu.cn (P.J.)

**Abstract:** Automated vehicle (AVs) technology is advancing at a rapid pace, offering new options for school travel. Parents play a decisive role in the choice of their child's school travel mode. To enable primary and secondary school students to take AVs to and from school, it is necessary to understand the factors that affect parents' intentions toward the new school travel mode. This study has three primary aims: (1) Discovering parents' intentions to escort children by AV and their potential determinants. (2) Constructing the Hybrid Choice Model (HCM) to examine the effects of parents' socioeconomic attributes, psychological factors, and travel attributes on using AVs to escort their children. (3) Raising practical implications to accelerate AV applications in school travel. The findings suggested that knowledge of AVs is the most important factor influencing parental intentions. Perceived usefulness, attitude, and perceived risk had significant effects on parental intentions. The direct effects of public engagement and perceived ease of use on parental intentions were not significant. Finally, this research can provide decision-making support for the government to formulate measures to promote AV application in school travel.

**Keywords:** school travel; automated vehicles; parents' intention; travel mode choice behavior; hybrid choice model

## 1. Introduction

Children's travel to and from primary and secondary school, which is called school travel, is a tricky issue. The World Health Organization (WHO) estimated that about 180,000 children die in road traffic accidents yearly, ranking in the world's top causes of death among children aged 5–14 [1]. This high fatality rate has aroused widespread concerns about school travel safety, increasing parents' willingness to drive their children to and from school [2]. However, escorting children by cars seems to have the opposite effect. More than half of the traffic casualty rate in students every year occurs in students who go to school by car [3], leading to an increasingly severe safety problem in school travel. Moreover, driving children to and from school provides other social problems such as traffic congestion [4,5] and pickup difficulty [6].

Automated vehicle technology could become a new solution to the problems mentioned above. The development of automated vehicle technology is boosting school travel quality [7,8]. Fully automated vehicles could perform all driving tasks independently. Previous studies have been devoted to estimating the potential benefits of AVs, including travel safety improvement, human-caused traffic accident reduction, traveler mobility increase, and traffic congestion alleviation [8–13]. The potential benefits of AV usage can correspond to the problems faced by school travel. Low public acceptance would diminish the expected benefits of AVs [14–18]. An essential prerequisite for achieving the expected benefits is the general parental acceptance of using AVs to escort children to school. Therefore, it is necessary to identify the essential factors of parental intention to use AVs to escort their children to school before the official arrival of AVs.

Previous studies have attempted to understand parental perceptions toward using AVs to transport children to and from school. Studies reveal that parents with different socioeconomic attributes show controversial opinions about using AVs [7,8]. There is an undeniable fact that both observed variables and unobserved factors could affect an individual's travel mode choice behavior [19]. Jing et al. [20] illustrated the significant influence of psychological variables on parents' perceptions of transporting children to and from school by AV. Analysis only focusing on psychological variables ignores the difference in decision making caused by individual heterogeneity. An integrated model involving psychological and observed variables could be easily understood and could exhibit better predictive power [21]. Hence, we applied a more comprehensive model structure named the hybrid choice model (HCM) to draw a complete picture of parental perceptions of AVs. The integrated model allows presenting complex relations between unobserved factors and socioeconomic characteristics towards parental choice behavior.

This research contributes to the literature in the following aspects.

(1) We attempt to discover parents' intentions to escort children by AV, which could provide a robust theoretical basis for developing and promoting AV application in school travel.

(2) An integrated model is proposed to analyze both the effects of observed and unobserved variables on parents' choice behaviors regarding AVs, providing a new opinion on factor analyses of AV usage in school travel research.

(3) Practical implications are raised to accelerate AV applications in school travel and support the government to formulate measures to promote rapid market occupancy of AVs.

The remainder of this paper is organized as follows. Section 2 summarizes the literature background and proposes the research hypotheses. Then, we describe the questionnaire design and the survey process (Section 3). Section 4 presents the results of the structural equation model and hybrid choice model. The most important implications of the present study are discussed in Section 5, and concluding remarks are drawn in Section 6.

## 2. Literature Review

As declared in Section 1, our study contributes to the understanding of parents' intentions of choosing AVs to escort their children. This section therefore contains relevant background and method material. The first part is a brief review of the hybrid choice model and related models, and the second and third parts present the theoretical framework and hypotheses used in this study.

### 2.1. Extended Choice Modeling

The decision process has been the focus of many scholars [22–24] and travel mode choice is a complex decision process. People's choice behavior is not only affected by objective characteristics but also subjective variables. Some scholars have attempted to explain people's choice behavior from the perspective of objective variables. Chillón et al. (2014) [25] used generalized linear mixed models (GLMM) with log-link functions to explore how safety and weather influence children's active school travel. The results from the multinomial logistic models of Shokoohi et al. (2012) [26] showed that the numbers of cars in a household and household income are the two main moderators on children's school travel. An in-depth understanding of the role of psychological factors on travel mode choice and psychological variables have gradually been included to reveal the mechanism of their influence on decision-making [27–29]. Meanwhile, structural equation modeling (SEM) is a multivariate statistical technique to determine the relationship between exogenous and endogenous structures and effectively explore the relationship between latent variables. Jing et al. [20] applied the SEM model to analyze how psychological factors influenced parents' AVs usage intention in school travel. Mehdizadeh et al. [30] analyzed the direct and indirect effects of parental attitudes on school travel patterns based on SEM. However, there is a complex interaction between objective and psychological variables that together affect individuals' behavioral intentions [31]. Existing research focuses on only one objective variable or psychological factor, which can hardly reflect the complexity of parental choice



behavior. [32] introduced psychological variables into the discrete choice model to construct the HCM, considering the influence of subjective and objective variables. Research on the HCM illustrated that the integrated model could facilitate understanding of the relationship between the individual's intention and antecedent variables [33].

HCM has been widely applied to explore factors affecting people's travel mode choices [34–38]. Kim et al. [39] examined and estimated the effects of individual latent attitudes and activity–travel context on car-sharing decisions. Kamargianni and Poly-doropoulou [31] adopted HCM to investigate the influences of children's attitudes toward cycling and walking on their mode choice behavior. The above studies adopted HCM to find exciting conclusions and provide insights for targeted policy interventions and related departments. Employing HCM to identify subjective and objective variables that influence parents' intentions to escort their children by AV could be feasible. Therefore, this research aimed to integrate SEM results into HCM to comprehensively investigate the parental choice of using AVs to escort children.

### 2.2. Theoretical Framework

The Technology Acceptance Model (TAM), proposed by Davis (1989) [40], is one of the most popular theories for understanding consumer acceptance of innovative technologies or new products [41–43]. TAM is fundamentally based on the "belief–attitude–intention" approach and also proposes two internal beliefs: perceived usefulness (PU) and perceived ease of use (PEU). Psychological variables from the TAM have been applied to analyze travelers' choices and acceptance of travel modes. Wang et al. [44] explored the key factors influencing consumer adoption of electric vehicles based on TAM and found that perceived usefulness and attitudes have a positive impact on the adoption of EVs. Wu et al. [45] examined user acceptance of electric autonomous vehicles by using TAM. Subsequently, scholars critically argued that although the TAM could explain individual behavioral intention to some extent, there is still a necessity to further find the potential impact factors to improve the explanatory power [46,47].

Several scholars have identified perceived risk as one of the barriers that directly or indirectly affect the usage or adoption intention of Avs [48], or car-hailing services [49]. Therefore, perceived risk was incorporated into the extended TAM in this study. In addition, knowledge plays an important role in behavioral research, which is closely related to individual behavioral decisions. Previous studies showed that the more people that know about new technologies, the stronger their intention to use them [44,50]. Research related to behavioral intention indicates that people in different cultural contexts are often assigned specific cultural values which may influence their behavioral decisions [51,52]. Face consciousness, embedded in cultural values with Chinese characteristics, is yet to be tested regarding whether it will influence parents' behavioral decisions in choosing Avs to escort their children. In addition, public engagement has been reported to greatly influence people's intentions to accept and use new technologies [53,54]. Understanding the role of public engagement in parental choice behavior could further provide practical insights for using Avs for school travel. This study seeks to investigate the extension of the TAM with knowledge of Avs, perceived risk, public engagement, and face consciousness.

To sum up, we verified the relationship of "belief–attitude–intention" through the hypothetical direct and indirect paths contributing to the study of the intention and choice behavior of Avs for school travel.

### 2.3. Research Hypotheses

The original TAM is difficult to use to fully explain user choice and acceptance of emerging technologies [46,54–57]. We propose an extended TAM model, including parental intention, attitude, perceived usefulness, perceived ease of use, perceived risk, knowledge of AVs, public engagement, and face consciousness. Attitude is a key factor affecting individual intention. Zhang et al. [58] pointed out that if users have a positive attitude towards innovative technology, they are more likely to accept and use the technology.

Perceived usefulness is closely related to people's attitudes and intentions toward innovative technologies [59]. Zhang et al. [60] and Wang et al. [44] found that perceived usefulness could positively affect consumer attitudes and intentions to use innovative technologies. Perceived ease of use significantly affects a user's intention to use innovative technologies [56]. In addition, perceived ease of use positively affects attitudes and perceived usefulness [58,61].

**Hypothesis 1 (H1).** *Attitude positively impacts parents' intentions to escort children by AV.*

**Hypothesis 2 (H2).** *Perceived usefulness positively impacts parents' attitudes to escorting children by AV.*

**Hypothesis 3 (H3).** *Perceived usefulness positively impacts parents' intentions to escort children by AV.*

**Hypothesis 4 (H4).** *Perceived ease of use positively impacts perceived usefulness.*

**Hypothesis 5 (H5).** *Perceived ease of use positively impacts parents' attitudes to escort children by AV.*

**Hypothesis 6 (H6).** *Perceived ease of use positively impacts parents' intentions to escort children by AV.*

Perceived risk negatively affects users' attitudes and intentions to accept and use innovative technologies [48,51], as well as perceived usefulness. Brecht et al. [62] and Ward et al. [63] pointed out that users with higher knowledge of AVs have more positive use intentions, which could also positively impact perceived ease of use. When users have a high level of understanding of new technology, the perceived risk is generally lower, and they are more likely to recognize that the new technology could bring benefits [44]. Based on the above discussion, the hypotheses are as follows:

**Hypothesis 7 (H7).** *Perceived risk negatively impacts perceived usefulness.*

**Hypothesis 8 (H8).** *Perceived risk negatively impacts parents' attitudes toward escorting children by AV.*

**Hypothesis 9 (H9).** *Perceived risk negatively impacts parents' intentions to escort children by AV.*

**Hypothesis 10 (H10).** *Knowledge of AVs positively impacts parents' intentions to escort children by AV.*

**Hypothesis 11 (H11).** *Knowledge of AVs positively impacts perceived ease of use.*

**Hypothesis 12 (H12).** *Knowledge of AVs positively impacts perceived usefulness.*

**Hypothesis 13 (H13).** *Knowledge of AVs positively impacts perceived risk.*

Public engagement is the degree to which public members, including residents, public interest groups, business or professional associations, and government organizations, actively participate in decision making and express opinions [64]. Public engagement is an essential element of the decision-making process, and much research is exploring its role in user acceptance and innovative technologies [53]. Parkins et al. [53] found that users with higher engagement levels had better knowledge about new technologies and were more willing to adopt solar energy technologies. Furthermore, Wang et al. [54] and Mah et al. [65] found that public engagement can help users reduce the perceived risks of nuclear energy

technology and increase support for nuclear energy utilization and development. AVs are also an innovative technology, but, so far, few studies have dissected the role of public engagement in selecting and accepting AVs, so this study included public engagement. This study defines public engagement as a parent's assessment of how actively they participate in AV-related activities. According to relevant research conclusions, we can speculate that as parents become more active in AV-related activities, it may increase their knowledge of AVs and reduce the perceived risk of escorting children by AV, deepen the perception of the benefits of escorting children by AV and make them more willing to choose this new school travel mode. Based on the above analyses, the hypotheses are as follows:

**Hypothesis 14 (H14).** *Public engagement positively impacts knowledge of AVs.*

**Hypothesis 15 (H15).** *Public engagement positively impacts perceived risk.*

**Hypothesis 16 (H16).** *Public engagement positively impacts parents' intentions to escort children by AV.*

**Hypothesis 17 (H17).** *Public engagement positively impacts perceived ease of use.*

**Hypothesis 18 (H18).** *Public engagement positively impacts perceived usefulness.*

Face consciousness refers to the belief that people desire to have good social self-worth and be respected in social activities [66]. It is closely related to people's social status and prestige. Face consciousness could significantly impact behavioral decisions [60,67,68]. Qian and Yin [52] found that symbolism is crucial in applying and promoting new technology products. As one of the new technology products, AVs also have a strong symbolic meaning, which can show the user's status and self-image [69]. In this study, face consciousness reflects how parents believe that using AVs to escort their children will maintain or improve their self-image or social status. According to relevant research conclusions, we can speculate that under the influence of face consciousness, parents may not only focus on the benefits of using AVs to escort their children but may also want to demonstrate social status and self-image by adopting this new model of school travel. Based on the above analyses, the hypotheses are as follows:

**Hypothesis 19 (H19).** *Face consciousness positively impacts parents' intentions to escort children by AV.*

**Hypothesis 20 (H20).** *Face consciousness positively impacts perceived risk.*

Figure 1 shows the variable relationships of the research model in this study. The research model is developed based on the extended TAM and the dependent variable is the parental intention of AVs usage.

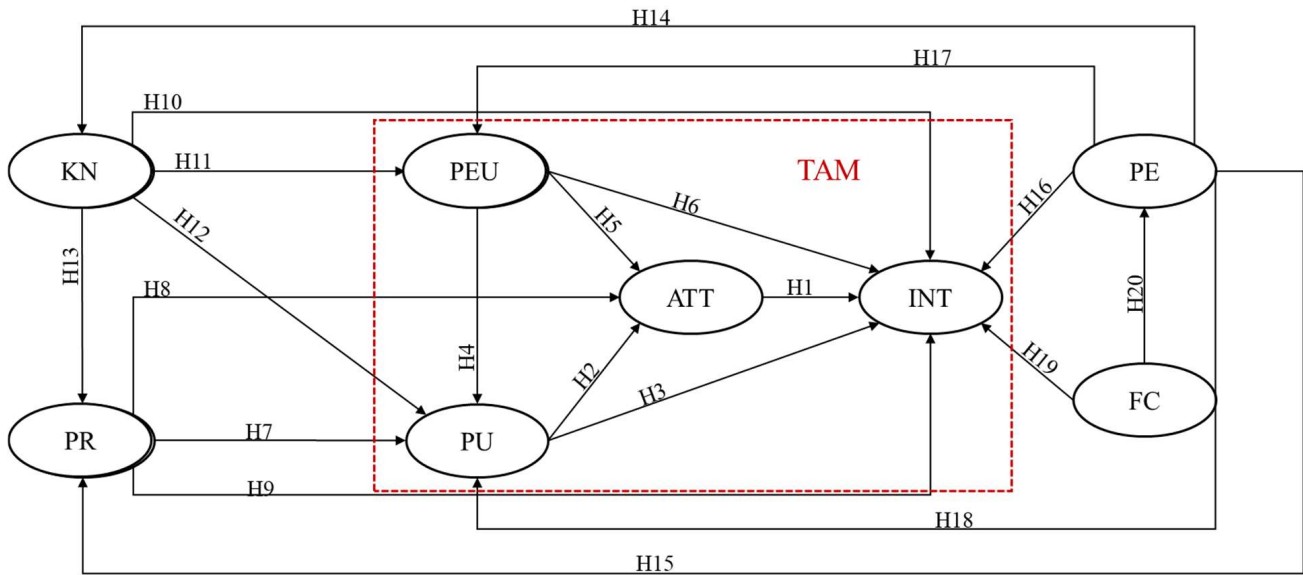

**Figure 1.** The proposed conceptual model. Note: INT = Intention; ATT = Attitude; PU = Perceived usefulness; PEU = Perceived ease of use; PE = Public engagement; FC = Face consciousness; KN = Knowledge of AVs; PR = Perceived risk.

### 3. Methods

This study aimed to assess how parents perceived escorting children by AV. We used questionnaires to collect the data for this study. The survey started on 5 September 2019 and ended on 22 September. Both holidays and weekdays were included in the survey. The survey was conducted in Jining, China. Questionnaires are randomly distributed to pedestrians. We interviewed 45 parents and revised the questionnaire based on their responses during the pre-survey. A total of 404 respondents completed the questionnaires in the official survey. We received approval from the relevant administrative departments and randomly recruited the respondents in places where people congregate (such as markets, plazas, and stations).

#### 3.1. Survey Design

The questionnaire is split into two parts. In the first part, the respondents were asked to answer questions relating to gender, age, income, education, and other demographic information. The second part consisted of items to measure psychological factors (such as using intention, attitude, perceived usefulness, etc.). Each factor is measured by three or four items. Items and reference sources are listed in Appendix A. Based on a five-point Likert scale, the level of respondents' agreement to the items was assessed, ranging from 1 to 5 (1 = totally disagree, 5 = totally agree) [70]. In addition, we designed three travel scenarios using combined pictures and characters to help respondents clearly understand the items to address the nonexistence of level 5 AVs. We made some modifications based on the scenarios we designed to make the items suit our research and randomly interviewed 45 parents to test the questionnaire in the pre-test. Figures 2–4 correspond to scenarios 1, 2, and 3, respectively. The figures show the critical information of different scenarios to aid in understanding the scenario's setting.

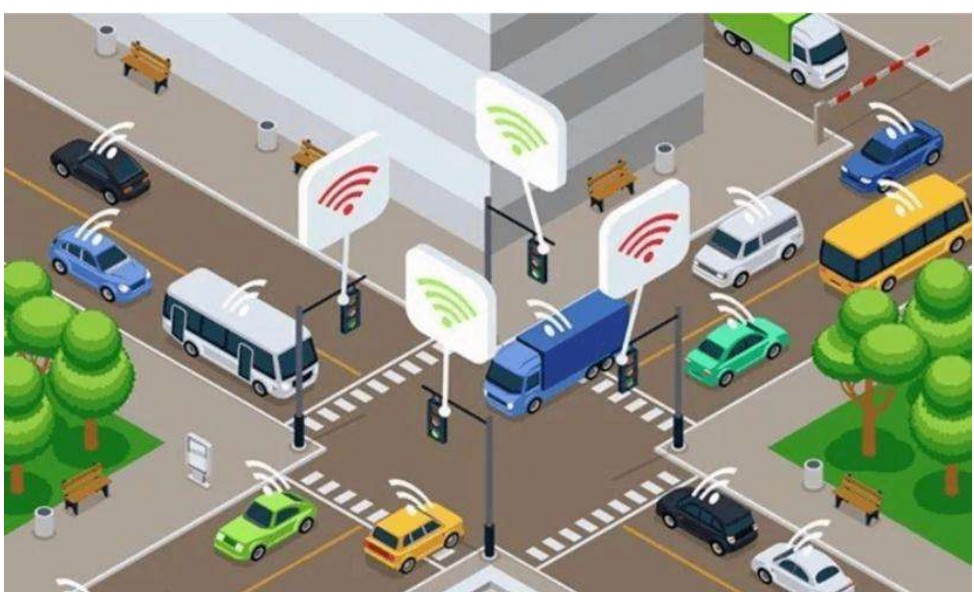

**Figure 2.** Usage scenario 1 of AVs.

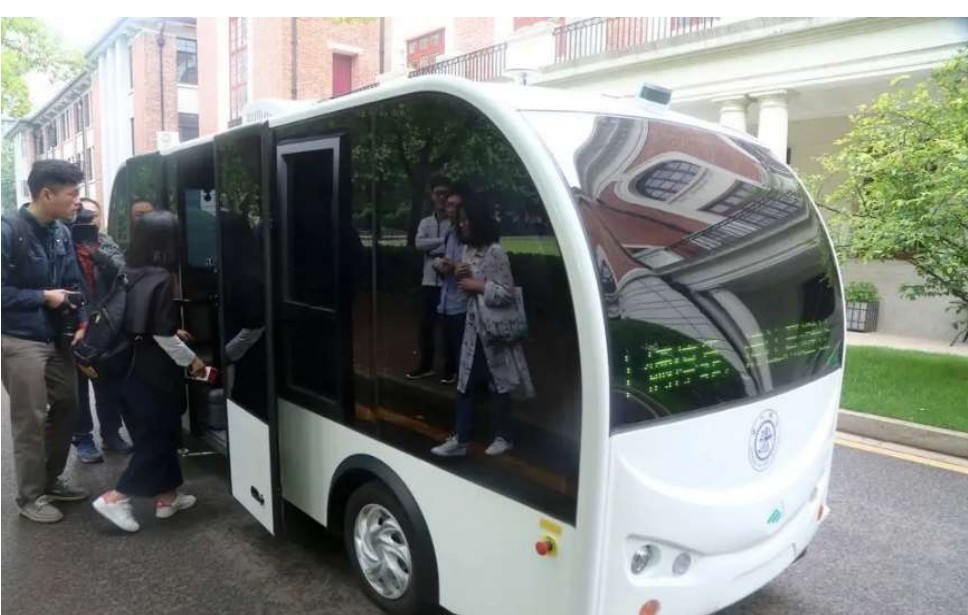

**Figure 3.** Usage scenario 2 of AVs.

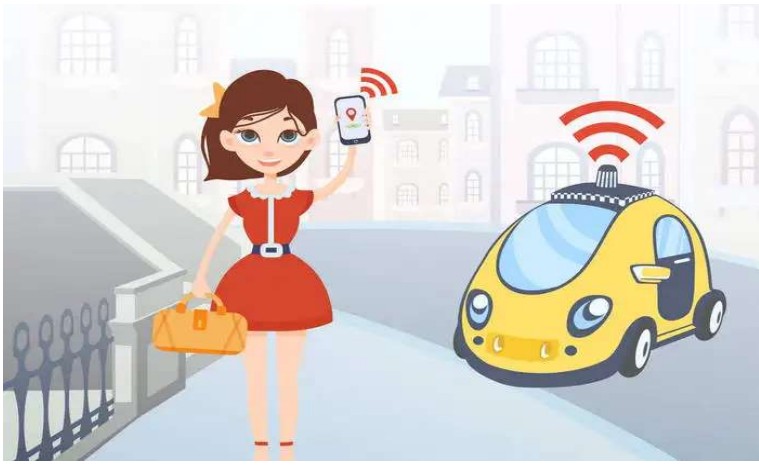

**Figure 4.** Usage scenario 3 of AVs.

Scenario 1 (see Figure 2): Level 5 AVs drive automatically and do not need a driver's manual operations. AVs keep a safe distance from other cars at an intersection by interacting using intelligent devices. They avoid pedestrians and other cars by themselves with a sensing system. AVs can complete operations such as overtaking, lane changing, distance keeping, speed adjustment, and other operations on the way.

Scenario 2 (see Figure 3): If you are so busy that you do not have time to escort your children, you could use the level 5 automatic vehicle to help you. You could keep children in the safety seats and set the destination by touching the screen or speaking. After that, you could continue your work or rest. The automatic vehicle will park at your designated location after the children leave the car.

Scenario 3 (see Figure 4): The automatic vehicle could adjust its speed based on the specified arrival time to ensure that your children would not be late. You could confirm the children's location by GPS. Communication equipment would help you communicate with your children. After a day of study, you could set the automatic vehicle to arrive at the school gate on time to pick up children from school.

### 3.2. Data Collection

A total of 404 respondents completed the questionnaires and 340 questionnaires were incorporated into our final analysis after filtering the questionnaires. A total of 59.12% of respondents were female and 40.88% were male. The main age distribution among the respondents was 31–40 (43.53%). More than half of respondents drive less than twice a week. A total of 49.71% of the respondents had a monthly income of more than CNY 4000 (USD 570). Most homes were 2 to 3 km away from school, accounting for 39.12% of respondents, followed by 3 to 4 km, accounting for 29.12%. Furthermore, we investigated the number of children in the family. The demographic information is shown in Table 1.

**Table 1.** Statistics of characteristics of samples (N = 340).

| Attributes | Variables | Items | Frequency | Percentage (%) |
|---|---|---|---|---|
| Socioeconomic attributes | Gender | Male | 139 | 40.88 |
| | | Female | 201 | 59.12 |
| | Age | 21–30 | 86 | 25.29 |
| | | 31–40 | 148 | 43.53 |
| | | >41 | 106 | 31.18 |
| | Driver license | Yes | 250 | 75.53 |
| | | No | 90 | 24.47 |
| | Driving frequency | Everyday | 84 | 24.71 |
| | | 5–6 times a week | 40 | 11.76 |
| | | 3–4 times a week | 25 | 7.35 |
| | | 1–2 times a week | 43 | 12.65 |
| | | Never | 148 | 43.53 |
| | Income (monthly) | CNY <2000 (less than USD 285) | 54 | 15.88 |
| | | CNY 2001–4000 (USD 285–570) | 117 | 34.41 |
| | | CNY 4001–6000 (USD 570–855) | 132 | 38.83 |
| | | Over CNY 6001 (over USD 855) | 37 | 10.88 |
| | Education | Junior high school and below | 27 | 7.94 |
| | | High school | 49 | 14.41 |
| | | Junior college | 109 | 32.06 |
| | | Bachelor's degree | 144 | 42.35 |
| | | Master's degree and above | 11 | 3.24 |
| | Number of children | 1 | 189 | 55.59 |
| | | 2 | 142 | 42.06 |
| | | 3 | 8 | 2.35 |

**Table 1.** *Cont.*

| Attributes | Variables | Items | Frequency | Percentage (%) |
|---|---|---|---|---|
| Travel attributes | Travel distance | Within 1 km | 0 | 0.00 |
| | | 1~2 km | 98 | 28.82 |
| | | 2~3 km | 133 | 39.12 |
| | | 3~4 km | 99 | 29.12 |
| | | More than 4 km | 10 | 2.94 |

Note: km = kilometer.

## 4. Results

### 4.1. Measurement Model

Before building the structural relationship model of the study variables, we conducted a data quality report, including its reliability and validity. Next, the quality of the structural model and the path hypothesis were tested.

#### 4.1.1. Reliability and Validity Analysis

Confirmatory factor analysis (CFA) was used to examine the reliability and validity of the measurement model. Before that, we conducted the KMO and Bartlett sphericity tests to ensure the CFA was feasible. KMO value was 0.960 (>0.7) and the significance of the Bartlett sphericity test was 0.000 (variables are independent of each other). We conducted CFA using principal component analysis (PCA). The rotation component matrix was used to examine if the items match the psychological variable based on maximum variance. The rotation component matrix needs to meet the absolute value of the factor loading of an observed variable on only one common factor greater than 0.5. The results mentioned above met these requirements.

We examined the internal consistency using Cronbach's alpha and composite reliability (CR) values [71]. The lowest Cronbach's alpha value was 0.765 (> 0.7) [72] and CR values ranged from 0.771 and 0.954 (> 0.6) [71]. The validity of a measurement is reflected by convergent validity and discriminant validity. The average variance extracted (AVE) and standardized factor loading were used to test the convergent validity. The lowest AVE was 0.530 and standardized factor loading was 0.644. The convergent validity is adequate when both values are greater than 0.5 [71]. The convergent validity test results are shown in Table 2. The criterion for discriminant validity is that the square root of AVE corresponding to each latent variable should be higher than the correlation coefficient between the variable and other variables. The results met the standard (see Table 3).

**Table 2.** Convergent validity test results.

| Variables | Items | Standardized Factor Loading | AVE | Cronbach's $\alpha$ | CR |
|---|---|---|---|---|---|
| AT | AT1 | 0.909 | 0.754 | 0.924 | 0.924 |
| | AT2 | 0.860 | | | |
| | AT3 | 0.854 | | | |
| | AT4 | 0.848 | | | |
| PU | PU1 | 0.923 | 0.853 | 0.945 | 0.946 |
| | PU2 | 0.929 | | | |
| | PU3 | 0.949 | | | |
| PEU | PEU1 | 0.893 | 0.862 | 0.949 | 0.949 |
| | PEU2 | 0.942 | | | |
| | PEU3 | 0.949 | | | |
| IN | IN1 | 0.921 | 0.809 | 0.927 | 0.927 |
| | IN2 | 0.878 | | | |
| | IN3 | 0.898 | | | |
| PR | PR1 | 0.900 | 0.756 | 0.953 | 0.954 |
| | PR2 | 0.936 | | | |
| | PR3 | 0.921 | | | |
| | PR4 | 0.907 | | | |

**Table 2.** *Cont.*

| Variables | Items | Standardized Factor Loading | AVE | Cronbach's α | CR |
|---|---|---|---|---|---|
| PE | PE1 | 0.872 | | | |
| | PE2 | 0.892 | 0.756 | 0.911 | 0.903 |
| | PE3 | 0.844 | | | |
| FC | FC1 | 0.644 | | | |
| | FC2 | 0.742 | 0.530 | 0.765 | 0.771 |
| | FC3 | 0.791 | | | |
| KN | KN1 | 0.908 | | | |
| | KN2 | 0.918 | 0.826 | 0.935 | 0.934 |
| | KN3 | 0.900 | | | |

**Table 3.** Discriminative validity test result.

| | KN | PE | PR | PU | PEU | AT | IN | FC |
|---|---|---|---|---|---|---|---|---|
| KN | **0.909** | | | | | | | |
| PE | 0.774 | **0.869** | | | | | | |
| PR | −0.750 | −0.694 | **0.916** | | | | | |
| PU | 0.805 | 0.772 | −0.733 | **0.924** | | | | |
| PEU | 0.800 | 0.755 | −0.695 | 0.783 | **0.928** | | | |
| AT | 0.757 | 0.736 | −0.701 | 0.779 | 0.764 | **0.868** | | |
| IN | 0.754 | 0.710 | −0.704 | 0.763 | 0.734 | 0.731 | **0.899** | |
| FC | 0.752 | 0.737 | −0.677 | 0.810 | 0.737 | 0.732 | 0.727 | **0.728** |

Note: The bold data on the diagonal are the square root of AVE; the data below the diagonal are correlations between latent variables.

### 4.1.2. Structure Model and Hypothesis Tests

The model fitting situation could be evaluated by chi squared/degree of freedom ($\emptyset^2/df$), the Normed Fit Index (NFI), the Tucker–Lewis Index (TLI), the Comparative Fit Index (CFI), the Root Mean Squared Error of Approximation (RMSEA), and the Goodness of Fit Index (GFI). Criteria and results of the structural equation model fitting degree evaluation are shown in Table 4. The revised model was tested for hypothesis paths and demonstrated an overall satisfactory fit.

**Table 4.** Criteria and results of the goodness of fit for the theoretical model.

| Fit Index | $\chi^2/df$ | RMSEA | NFI | CFI | TLI | GFI |
|---|---|---|---|---|---|---|
| Measured value | 1.484 | 0.038 | 0.957 | 0.985 | 0.983 | 0.914 |
| Standard value | $1 < \emptyset^2/df < 3$ | <0.050 | >0.900 | >0.900 | >0.900 | >0.900 |
| Adaptation judgment | Yes | Yes | Yes | Yes | Yes | Yes |

Figure 5 shows the model's hypothesis test results and path coefficients, where the solid lines represent significant paths and the dashed lines represent the non-significant paths. The results showed that the attitude toward escorting children by AV affected parents' intentions positively, which supported H1. PU had a positive effect on attitude towards school travel and parents' intention, verifying H2 and H3. Furthermore, perceived ease of use positively affected perceived usefulness and attitude, supporting H4 and H5. Contrary to expectations, perceived ease of use had no significant effect on parents' intention, which was inconsistent with hypothesis H6. Perceived risk showed a negative effect on parents' intentions, attitudes, and perceived usefulness. Hence, H7, H8, and H9 were all supported. Among the proposed hypotheses related to knowledge of AVs, this could positively impact parents' intentions, perceived ease of use, and perceived usefulness, while negatively impacting perceived risk. Therefore, H10, H11, H12, and H13 were supported. The impact of public engagement on knowledge of AVs and perceived risk was significant, thereby verifying H14 and H15. Unexpectedly, public engagement had no significant effect on parents' intentions. Therefore, H16 was not supported. Public engagement had significant

positive impacts on perceived ease of use and perceived usefulness, supporting H17 and H18. In terms of face consciousness, the results showed that face consciousness could positively impact public engagement and parents' intention to use AVs, which confirms H19 and H20. In conclusion, except for H6 (perceived ease of use→parents' intention) and H16 (public engagement→parents' intention), the other hypothesized path relationships were significant. The standardized path coefficients and significance levels of the paths are shown in Table 5.

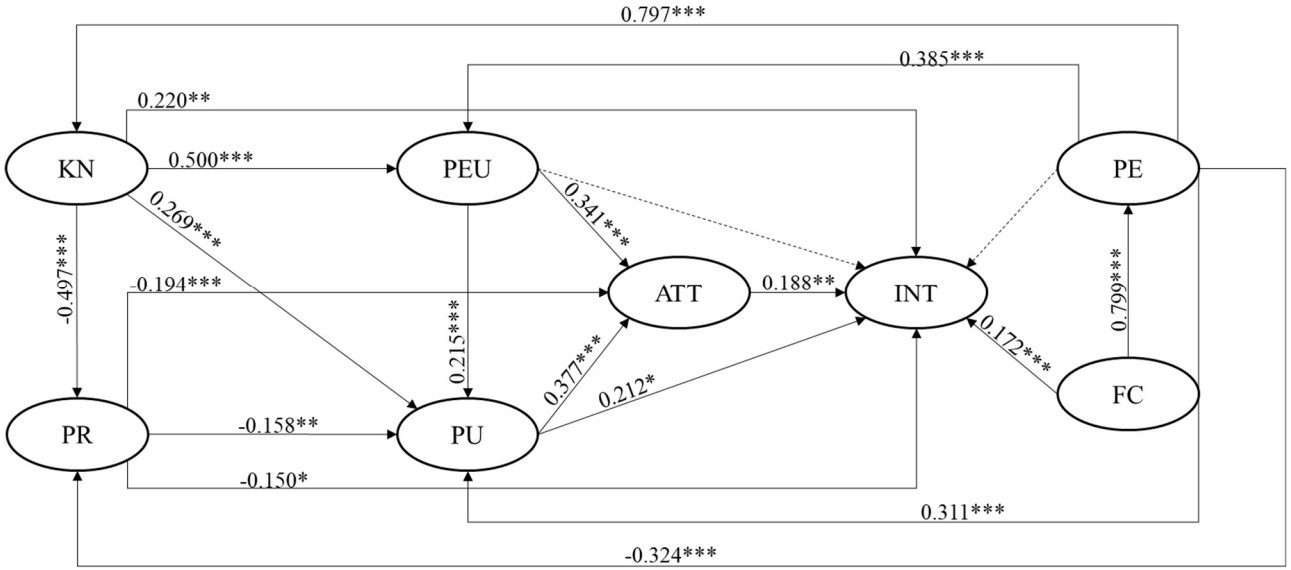

**Figure 5.** Path test of the structural equation model. Note: *** $p < 0.001$, ** $p < 0.01$, * $p < 0.05$.

**Table 5.** Standardized path coefficients and the significance level of the paths.

| Hypotheses | Path | Standardized Estimate | $p$ | Supported ($p < 0.05$) |
|---|---|---|---|---|
| H1 | AT→Intention | 0.188 | ** | Yes |
| H2 | PU→AT | 0.377 | *** | Yes |
| H3 | PU→Intention | 0.212 | * | Yes |
| H4 | PEU→PU | 0.215 | *** | Yes |
| H5 | PEU→AT | 0.341 | *** | Yes |
| H6 | PEU→Intention | 0.126 | 0.108 | No |
| H7 | PR→PU | −0.158 | ** | Yes |
| H8 | PR→AT | −0.194 | *** | Yes |
| H9 | PR→Intention | −0.150 | * | Yes |
| H10 | Knowledge→Intention | 0.220 | ** | Yes |
| H11 | Knowledge→PEU | 0.500 | *** | Yes |
| H12 | Knowledge→PU | 0.269 | *** | Yes |
| H13 | Knowledge→PR | −0.497 | *** | Yes |
| H14 | PE→Knowledge | 0.797 | *** | Yes |
| H15 | PE→PR | −0.324 | *** | Yes |
| H16 | PE→Intention | 0.036 | 0.747 | No |
| H17 | PE→PEU | 0.385 | *** | Yes |
| H18 | PE→PU | 0.311 | *** | Yes |
| H19 | FC→Intention | 0.172 | ** | Yes |
| H20 | FC→PE | 0.799 | *** | Yes |

Note: *** $p < 0.001$, ** $p < 0.01$, * $p < 0.05$.

### 4.2. Hybrid Choice Model

We integrated SEM results into the HCM to identify subjective and objective variables that influence parents' intentions to escort their children using AVs. The HCM framework is shown in Figure 6.

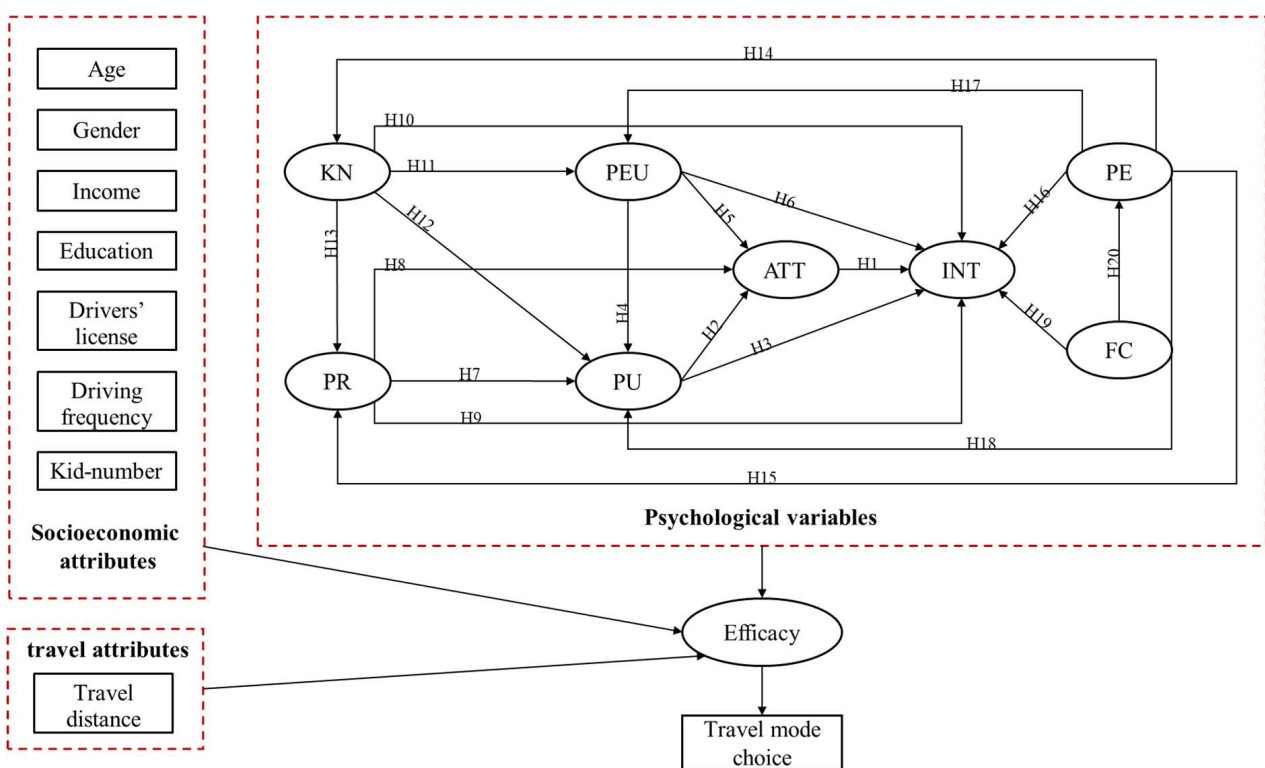

**Figure 6.** Hybrid choice model of parents' travel mode choice behavior.

Table 6 presents the goodness of fit of the HCM. We performed a Hosmer–Lemeshow test to confirm that the HCM had better goodness of fit. The index of Cox and Snell $R^2$ and Nagelkerke $R^2$ were 0.527 and 0.706, respectively. The model's overall accuracy was 85.3% and it could accurately predict parents' choice behavior. In conclusion, the model fits well.

**Table 6.** The goodness of fit of the hybrid choice model of parents' choice behavior.

| Evaluation Index | Results | Adaptation Judgment |
| --- | --- | --- |
| Hosmer–Lemeshow test | $p > 0.05$ | |
| Cox and Snell $R^2$ | 0.527 | |
| Nagelkerke $R^2$ | 0.706 | Fit Well |
| Overall accuracy | 85.3% | |

Parameter estimation results of the HCM are reported in Table 7. The results indicated that nine variables could significantly affect parents' choice behavior. As expected, gender was positively associated with parents' choice behaviors ($\beta = 0.711$, $p < 0.05$), which suggested that males are more likely to escort their children using AVs. Anania et al. [7] found that male parents in the US and India were more likely to escort their children using AVs, which is consistent with our findings. Age also played a role in mode choice ($\beta = 0.432$, $p < 0.05$). The distance was also a significant variable ($\beta = 0.528$, $p < 0.05$). Parents tended to use AVs more to escort their children as distance increased.

Regarding the psychological variables, results revealed that parent intention is significantly positively associated with their choice behavior ($\beta = 0.727$, $p < 0.05$). Parent choice behavior was demonstrated to be positively affected by knowledge of AVs ($\beta = 0.700$, $p < 0.05$) and negatively influenced by PR ($\beta = -0.551$, $p < 0.05$). Parents with a high-level knowledge of AVs might more easily realize the benefits of using AVs and reduce the PR caused by unfamiliarity with AVs. They could better weigh the advantages and disadvantages of using AVs and choose the new school travel mode to escort their children when they learn more about AVs. PU ($\beta = 0.798$, $p < 0.05$) and attitudes toward school travel to AVs ($\beta = 0.792$, $p < 0.05$) had significant positive effects on parents' choice behavior.

Parents' perceptions of the benefits of escorting children by AV could promote positive attitudes toward this new school travel mode and further encourage the use of AVs. Face consciousness was significantly positively related to parent choice behavior ($\beta = 0.879$, $p < 0.05$). Parents who value face consciousness deemed that adopting AVs could maintain or improve their social status and self-image. They preferred to choose the new school travel mode to escort their children in the future.

**Table 7.** Parameter estimation results of the hybrid choice model.

| Variables | Coefficient | *p* | Odds |
|---|---|---|---|
| Gender | 0.711 | * | 2.036 |
| Age | 0.432 | * | 1.541 |
| Distance | 0.528 | * | 1.695 |
| IN | 0.727 | * | 2.070 |
| KN | 0.700 | * | 2.013 |
| PU | 0.798 | * | 2.221 |
| AT | 0.792 | * | 2.208 |
| FC | 0.879 | * | 2.408 |
| PR | −0.551 | * | 0.576 |
| Cons | −14.743 | *** | 0.011 |

Note: *** $p < 0.001$, * $p < 0.05$.

A comparison analysis between a binomial choice model and the hybrid choice model could show the effectiveness of the proposed model. We used a logistic model to reveal parental choice behavior towards using automated vehicles to escort children, and made a comparative analysis of the logistic model and the hybrid choice model (see Table 8).

**Table 8.** Comparative analysis of logistic model and hybrid choice model.

| Variables and Goodness of Fit Evaluation Index | | Logistic Model | | Hybrid Choice Model | |
|---|---|---|---|---|---|
| | | Coefficient | *p* | Coefficient | *p* |
| | Gender | 0.953 | *** | 0.711 | * |
| | Age | 0.479 | ** | 0.432 | * |
| | Distance | 0.923 | *** | 0.528 | * |
| | IN | | | 0.727 | * |
| | KN | | | 0.700 | * |
| Variables | PU | | | 0.798 | * |
| | AT | | | 0.792 | * |
| | FC | | | 0.879 | * |
| | PR | | | −0.551 | * |
| | cons | −4.536 | *** | −14.743 | *** |
| Goodness of fit evaluation | Cox and Snell $R^2$ | 0.192 | | 0.527 | |
| | Nagelkerke $R^2$ | 0.258 | | 0.706 | |
| | Overall accuracy | 68.8% | | 85.3% | |

Note: *** $p < 0.001$, ** $p < 0.01$, * $p < 0.05$.

## 5. Discussion

### 5.1. Theoretical Implications

This study employed SEM to explore the psychological determinants of parental intention. We also integrated SEM results into the HCM to identify factors significantly influencing parental choice behavior. This section illustrates the theoretical implications based on the SEM and HCM results, respectively. Our findings provide insight for formulating relevant school strategies, contributing to the promotion of AVs for school travel.

5.1.1. The Interaction Effects among Latent Variables

This study gave an antecedent structure of parents' intentions to escort their children using AVs. The framework investigated the role of psychological variables in affecting AVs

usage intention. We also analyzed in detail the interaction between latent variables in the process of parental choice behavior.

Previous studies indicated that attitude significantly affected travelers' behavioral intentions [44,58,73]. Our study confirmed that attitude toward school travel using AVs had a significant positive impact on parents' intentions ($\beta = 0.188$, $p < 0.01$). This means that parents who have a positive attitude toward AVs are more likely to choose the new travel mode to transport their children to and from school. However, PR was demonstrated to be significantly negatively related to attitude towards school travel in AVs ($\beta = -0.194$, $p < 0.001$), PU ($\beta = -0.158$, $p < 0.01$), and parents' intentions ($\beta = -0.150$, $p < 0.05$). These findings are consistent with Jing et al. [49] that PR might make people feel negative about AVs and prevent them from using AVs to escort their children. Contrary to Liu et al. (2019), they found that PR did not significantly impact user intention to use AVs. The impact of perceived risk on AV usage intention varies by population and usage scenarios. When parents choose to use AVs to take their children to and from school, they are more concerned about the risks on the road. Knowledge of AVs positively affects parents' intentions ($\beta = 0.220$, $p < 0.01$), PU ($\beta = 0.269$, $p < 0.001$), and PEU ($\beta = 0.500$, $p < 0.001$), which confirms previous research [44,62]. The results of this study indicated that knowledge of AVs was a significant antecedent of perceived risk, and significantly mitigated parents' perceived risk regarding using AVs to escort their children ($\beta = -0.497$, $p < 0.001$). Furthermore, our findings showed that the relationship between PEU and parents' intentions was not significant. Moták et al. [74] and Xu et al. [17] suggested that PEU could not predict intention when respondents have no direct experience of AVs. Interestingly, we demonstrated a significant relationship between face consciousness and public engagement ($\beta = 0.799$, $p < 0.001$). Under the influence of face consciousness, people may actively participate in AVs activities, being reluctant to lag behind others when they see that others have the opportunity to participate.

The results highlight the direct significant impact of ATT, PU, PR, FC, and KN on parental intention of AV usage while PE and PEU had indirect effects. Based on the extended TAM, the complex interaction between the psychological variables in the process of parental intention was revealed. Furthermore, we demonstrated the applicability of the extended psychological variables in the study of parents choosing AVs to escort children.

### 5.1.2. The Mixed-Effects of Objective and Subjective Variables

Individuals' behavioral decisions are influenced by a combination of objective and psychological variables [31]. We constructed the HCM to examine the effects of parents' socioeconomic attributes, psychological factors, and travel attributes on using AVs to escort their children, and to provide a theoretical basis for promoting the application of AVs in school travel mode choice.

### Observable Variables

Observable variables in this study include gender, age, and travel distance. This study finds that parental choice behavior is significantly affected by gender ($\beta = 0.711$, $p < 0.05$). Men are 2.036-times more likely than women to choose AVs to escort their children (odds: 2.036). This is consistent with Anania's [7] findings. Age also significantly influences parental choice behavior ($\beta = 0.432$, $p < 0.05$). For every one-year increase in age, the probability of using AVs to escort their children is 0.541-times higher than other school travel modes (odds: 1.541). Travel distance significantly affected parents' choice behaviors ($\beta = 0.528$, $p < 0.05$). For every additional kilometer of distance, the probability of using AVs to escort their children is 0.695-times higher than other school travel modes (odds: 1.695). When travel distances get longer, it is more convenient to escort children by motor vehicles, so parents are willing to escort their children using AVs.

Subjective Variables

Psychological variables in this study included use intention, knowledge of AVs, perceived usefulness, attitude, face consciousness, and perceived risk. This study finds that use intention could significantly influence parental choice behavior ($\beta = 0.727$, $p < 0.05$). For each additional unit of usage intent, the probability of using AVs to escort their children is 1.070-times higher than other school travel modes (odds: 2.070). Knowledge of AVs could significantly affect parents' choice behaviors ($\beta = 0.700$, $p < 0.05$). For every one-unit increase in knowledge of AVs, the probability of using AVs to escort their children is 1.013-times higher than other school travel modes (odds: 2.013). Perceived usefulness also significantly impacts parents' choice behaviors ($\beta = 0.798$, $p < 0.05$). For every one-unit increase in perceived usefulness, the probability of using AVs to escort their children is 1.221-times higher than other school travel modes (odds: 2.221). Attitude is closely related to parents' choice behaviors ($\beta = 0.792$, $p < 0.05$). For every one-unit increase in attitude, the probability of using AVs to escort their children is 1.208-times higher than other school travel modes (odds: 2.208). Face consciousness could significantly affect parents' choice behaviors ($\beta = 0.879$, $p < 0.05$). For every one-unit increase in face consciousness, the probability of using AVs to escort their children is 1.408-times higher than other school travel modes (odds: 2.408). Perceived risk could significantly influence parental choice behavior ($\beta = 0.551$, $p < 0.05$), but unlike other psychological variables, perceived risk could hinder parents from choosing AVs to escort their children. For every one-unit increase in perceived risk, the probability of using AVs to escort their children is reduced 0.576-times that of using other school travel modes (odds: 0.576).

The value of the HCM seemed to be clear in estimation performance and the HCM provided an attractive improvement in modeling parents' mode choice behaviors. It provided insight into the importance of unobservable variables and objective attributes to mode choice, suggesting that the HCM is a powerful tool to improve the understanding of travel mode behavior to be used by relevant researchers, policy makers and manufacturers.

### 5.2. Practical Implications

From the exploration of using AVs to escort children, it appeared that attitude and perceived usefulness, perceived risk, knowledge of AVs, and face consciousness had a greater impact on parents' intentions. The results of the study provided insights into policies and campaigns to be used when AVs are applied in school travel in the future. Next, the practical implications are further elaborated on regarding four aspects for future AV manufacturers and the government.

### 5.2.1. Attitude and Perceived Usefulness

Relevant governments and automated vehicle manufacturers are considering multiple measures to improve the parental assessment of AVs. Manufacturers and the traffic management agencies could conduct campaigns targeting the utility, characteristics, and performance of AVs so that parents could better perceive the benefits of using AVs to escort their children to and from school. For example, using AVs may improve the time utilization of parents (no longer paying the time cost for picking up their children and using the time saved to do other things). Next, manufacturers could further facilitate dialogue and communication with users. By holding hands-on experience activities, users can be invited to test ride AVs, to feel the high-quality service brought by AVs and improve users' attitudes toward the new school travel mode.

### 5.2.2. Perceived Risk

Our study indicated that perceived risk is a negative factor for parents to choose AVs to escort children, and it is necessary to take measures to reduce the parental perceived risk. Based on the study results, we analyzed risk from the following three perspectives: functional risk, physical risk, and privacy risk. Firstly, manufacturers are recommended to issue statements explaining to users how AVs effectively protect passengers in the

event of an emergency to reduce parents' concerns about the risks of AV functionality. Manufacturers must also take the initiative to provide regular vehicle inspection services and strictly enforce safety testing to eliminate safety hazards. Secondly, parents are also worried about the physical risk of children riding in AVs and functional risks brought by the new technology. Parents could accomplish monitoring the conditions of their children in the car using interior cameras and interactions between mobile devices and AVs. When an abnormal situation is detected, the vehicle could be controlled to stop and issue an alarm through the mobile phone, and at the same time, passersby or the police could be asked for help. However, interior cameras in AVs could raise privacy risk concerns for parents. On the one hand, it is recommended that the government introduces policies and measures for the management of information captured by AVs and protects user privacy in the form of laws. On the other hand, the government can step up regulation of manufacturers so as to prevent information from leaking or being illegally used, reducing parents' concerns about privacy risks. In addition, the government could strengthen cooperation with relevant scholars, the insurance industry, and automated vehicle manufacturers, ensuring the applicability and authority of the autonomous driving safety code of conduct.

### 5.2.3. Knowledge of AVs

The results showed that improving parents' knowledge of AVs effectively promotes AV application in school travel. Manufacturers may consider making AV science information brochures for distribution. Traffic management agencies could cooperate with manufacturers to hold AV scientific education activities, such as professional lectures and technical exhibitions, to provide multiple channels for parents to learn about autonomous driving technology. In addition, through magazines, newspapers, TV programs, or other public media, businesses can subconsciously increase the public knowledge of AVs.

### 5.2.4. Face Consciousness

Parents are not only concerned about the benefits brought by using AVs to escort their children but also want to show social status and self-image by adopting the new mode of school travel. The bodywork of AVs could be equipped with distinctive logos to enhance identifiability and uniqueness, and thus indirectly demonstrate the social status of the user. Consumer-targeted advertising implies face-related features that underline the ability of AVs to bring social prestige. In addition, consumers in collectivist cultures place a high value on social identity [75]. AV usage as a potential strategy to help consumers gain importance and recognition among groups could be highlighted. A previous study has also shown that consumers with a higher face consciousness value the fashion and novelty of products [76]. Therefore, manufacturers may also pay attention to the visual design of AVs and develop more fashionable and novel shapes to meet the needs of different consumers to express their self-image.

## 6. Conclusions

The development of AV technology has made it possible for parents to choose AVs to escort their children to and from school. Before the arrival of AVs, understanding parental intention and impact factors is necessary. We conducted empirical research on parents' choice behaviors of escorting children using AVs. This research provides an early step in the study of parental intention. Furthermore, face consciousness and public engagement were introduced into the field of AV acceptance, which expanded the consideration of psychological factors. An integrated model was used to analyze the effects of observed and unobserved variables on parents' choice behaviors, providing new insights into factor analyses of AV usage in school travel research. The main findings are as follows. We explored the effect of psychological variables on parental intention to use AVs based on SEM. The results showed that face consciousness could influence parents' intentions positively. The knowledge of AVs and public engagement are important factors influencing perceived risk and could reduce the negative effect of perceived risk on intention. In addition,

we used the HCM and introduced socio-economic attributes and travel attributes to explore parents' choice behaviors. The results indicated that gender, age, and travel distance could affect parents' intentions significantly, which can help manufacturers design and sell specific schemes. Moreover, the positive influence of perceived risk, perceived usefulness, and attitude supplements our understanding of parents' intentions.

## 7. Research Prospects and Limitations

We explained parents' behaviors using AVs to escort children. However, this study could be improved with regard to its limitations. The first limitation is that most respondents have no AV experience. The discussion is based on respondents' initial perceptions of their knowledge about AVs gained from the Internet or other channels. Parents' intentions to escort children using AVs may change with the popularity of the new technology. A longitudinal tracking study could be recommended to deeply understand the evolution of parents' choice behaviors in the future. The second limitation is that our sample only comprised Jining survey data. Future studies could explore parents' intentions across provinces or in different countries. The third limitation is that most respondents are younger and with more education. This case may limit sampling and respondent availability. This deficiency could be made up through larger-scale investigations. Moreover, emergencies that occur when a child travels alone in an AV to school will be a factor to consider in subsequent studies, such as if the child suddenly becomes ill on the way, the AV breaks down or cannot negotiate a particular scenario, etc. These are some of the factors which would need to be addressed before most parents would be comfortable using AVs to escort their young children to school. In further research, we will take these factors into account. Finally, service levels of current school travel modes could become another critical factor of parental choice behavior. Future research could construct group analyses based on current school travel modes.

**Author Contributions:** Conceptualization, Y.M. and Q.M.; methodology, Y.M. and Q.M.; software, P.J. and Y.Z.; validation, Y.X.; formal analysis Y.Z. and Y.X.; investigation, J.H. and D.S.; resources, P.J.; data curation, P.L.; writing—original draft preparation, Y.M.; writing—review and editing, Q.M.; visualization, Y.X.; supervision, Q.M.; project administration, Y.M.; funding acquisition, P.J. All authors have read and agreed to the published version of the manuscript.

**Funding:** This work was supported by the National Natural Science Foundation of China (grant number 71871107).

**Institutional Review Board Statement:** Not applicable.

**Informed Consent Statement:** Informed consent was obtained from all subjects involved in the study.

**Data Availability Statement:** Not applicable.

**Acknowledgments:** The authors gratefully acknowledge the kind support from the National Natural Science Foundation of China. We are also grateful to colleagues who provided valuable comments during the paper writing process.

**Conflicts of Interest:** On behalf of all authors, the corresponding author states that there is no conflict of interest.

## Appendix A

**Table A1.** Sources of constructs and items used in the research.

| Constructs | Item | Source |
|---|---|---|
| Knowledge of AVs (KN) | KN1: I understand the performances of AVs (such as during driving, drivers are not necessary). <br> KN2: I understand the potential risk used AVs (such as AVs may be hacked). <br> KN3: I understand the advantages of AVs (such as reducing crashes by a human). | [44] |
| Public Engagement (PE) | PE1: I am willing to concern proactive with some information about AVs. <br> PE2: I am willing to participate in training activities about AVs offered by the government or manufacturer if I have the opportunity. <br> PE3: I am willing to participate in AVs conferences if I can. | [77] |

**Table A1.** *Cont.*

| Constructs | Item | Source |
|---|---|---|
| Perceived Risk (PR) | PR1: I am worried that AVs will divulge family privacy because of Internet security.<br>PR2: I am worried that AVs will break down when picking up children to and from school.<br>PR3: I am worried that AVs could not adapt to bad weather and rugged terrain.<br>PR4: I am worried about the safety of children in the AVs. | [58,78] |
| Face Consciousness (FC) | FC1: Picking up children to and from school using AVs will bring me prestige.<br>FC2: If my relatives and friends use AVs to pick up children to and from school, I will do too.<br>FC3: I hope to gain recognition from my relatives and friends for using AVs to pick up children to and from school. | [75,76] |
| Attitude (ATT) | AT1: I am very interested in using AVs to pick up children to and from school.<br>AT2: I support that children are picked up to and from school using AVs.<br>AT3: I think using AVs to pick up children to and from school is a good idea.<br>AT4: I think using AVs to pick up children to and from school is feasible. | [58,59] |
| Perceived Usefulness (PU) | PU1: Using AVs to pick up children to and from school is safer by reducing crashes caused by humans.<br>PU2: Using AVs to pick up children to and from school could save my time.<br>PU3: Using AVs to pick up children to and from school is better for traffic by reducing congestion. | [59] |
| Perceived Ease of Use (PEU) | PEU1: I think AVs are easy to learn.<br>PEU2: I think it is easy for me to learn to control AVs for picking up children to and from school.<br>PEU3: I think children could ride an AV easily and leave school with enough practice. | [59] |
| Intention (INT) | IN1: I will buy an AV after AVs hit the market.<br>IN2: I will use AVs to pick up children to and from school after I possess an AV.<br>IN3: I will recommend my relatives and friends to pick up children to and from school using AVs. | [16] |

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
