# Peer review of "Factors Affecting the Parental Intention of Using AVs to Escort Children: An Integrated SEM–Hybrid Choice Model Approach"

_sustainability, doi:10.3390/su141811640_

Round 1

Reviewer 1 Report

This study used a hybrid choice model to analyze the parents’ perception of using automated vehicles for school travel. The manuscripts are well-organized and well-written. The results were compared with existing studies, and limitations were also discussed. I have a few suggestions for possible improvements.

 Major

1.      Please elaborate on the questionnaire survey. When is the survey date? Was it on holiday or weekday? Were the targets only the parents with children? You mentioned 45 parents on page 6, but you also mentioned 404 respondents on page 7. The 45 parents are only for pretest and presurvey?

2.      Table 1: Please explain the “travel distance.” I suppose it was the distance between home and school. Was the unit “kilometer?” 

3.      Please explain the “travel mode choice” in your model. Was it a binary choice? The choice between AVs and the current mode? How do you consider the service level of the current mode?

 4.      Comparing the results of the standard discrete choice model and your hybrid choice model might be useful to show the effectiveness of your proposed framework.

Minor

1.      Please define the abbreviation properly. In the abstract, the definitions of “AVs” and “HCM” are missing. In the main text, “HCM” is defined twice (p.2 and 3).

2.      References to Figures 2, 3, and 4 appear missing.

3.      The authors’ affiliations information is missing. 

Reviewer 2 Report

It's an interesting topic to study the intention of using AVs to escort children in school travel. There are some doubts in the manuscript.

1. 2.3 Research hypotheses The first paragraph of the hypothesis seems illogical. It is suggested that the hypothesis of the factors in TAM could be stated first, then the hypothesis of other factors, and not to mix them. In addition, the original definition of the variables could be argued first, then applications in existing studies, and finally the definition of the variables in this study.

2.  3.2 Data collection The authors provide the group proportion of the respondents’ travel distance between home and school, but the group name cannot directly reflect the distance range. Please provide the detailed value of the distance range.

3.  6 Conclusion The innovations of this study are not presented again in the conclusion part. The current description is only a summary of this study and a partial presentation of the results, which is not attractive enough. It is recommended that the authors reorganize the conclusion part.

Reviewer 3 Report

This submission addresses parents’ willingness to use AVs to escort children to and from school, a topic which may increase in importance as technology advances. The submission is reasonably well written, but it could benefit from a thorough review by a skilled technical editor. I have included just a few examples of my concerns with the writing as well as other concerns below:

1.       P. 3 - “Studies showed that people know more enough about new technologies, they increase their usage intention”

a.       This sentence must be rewritten; I cannot interpret as written.

2.       P. 4 - “To sum up, we used to verify the relationship of “belief-attitude-intention” through the hypothetical direct and indirect paths contributing to the study of the intention and choice behavior of AVs for school travel.”

a.       Appears to be a missing word between “used” and “to”

3.       P. 6 – “Based on the five-point Likert scale, the level of respondents agree to the items is assessed, ranging from 1 to 5 (1= totally disagree, 5= totally agree)”

a.       “respondents agree” should be “respondents’ agreement”

4.       P. 7 – “Scenario 3: The automatic vehicle could adjust speed based on the specified arrival time to ensure that your children would not be late.”

a.       This implies the AV would speed if necessary to prevent the child from arriving to school late… this sounds like a decrement in safety, not an enhancement

5.       P. 7 – “. You could confirm the children’s location by GPS.”

a.       The parent can easily accomplish this in a variety of ways w/o an AV

6.       Bigger problem is there’s no basis in this study to suggest that the government should be manipulating people to use AVs as escorts

7.       This study ignores the idea of safety apart from automotive safety – that is, who’s in the car w/ the kids? Bad guys? Who’s there to protect them? What do the kids do if the AV breaks down or cannot negotiate a particular scenario? What if the kid needs to use the restroom or is sick or whatever?  Can the AV handle that? These are just some of the factors which would need to be addressed before most parents would be comfortable using AVs to escort their young children to school.

Round 2

Reviewer 1 Report

Thank you for your revision. It is satisfactory.

However, please check Table 8 again.  The variable "Gender" appears twice.

Reviewer 3 Report

the authors have addressed my concerns to a satisfactory degree

Author Response

Dear Reviewer 3:

Thanks again for your letter concerning our manuscript entitled “Factors affecting the parental intention of using AVs to escort children: An integrated SEM-Hybrid Choice Model approach”. We want to express our gratitude to reviewer 3 for the feedback and we are so pleased that you approve our revision of the manuscript. We are grateful to have the comments and guidance given by reviewer 3 that made our manuscript more complete and meaningful. Regarding the problems pointed out by the reviewers, we do our best to correct them. We invite experts to help us revise the English language and we will also strive for further improvements and enhancements to the writing and research objectives of this study.